# *Hymenolepis diminuta* Reduce Lactic Acid Bacterial Load and Induce Dysbiosis in the Early Infection of the Probiotic Colonization of Swiss Albino Rat

**DOI:** 10.3390/microorganisms10122328

**Published:** 2022-11-24

**Authors:** Sudeshna Mandal, Chandrani Mondal, Tanmoy Mukherjee, Samiparna Saha, Anirban Kundu, Sinchan Ghosh, Larisha M. Lyndem

**Affiliations:** 1Parasitology Research Laboratory, Department of Zoology, Visva-Bharati University, Santiniketan 731235, India; 2Department of Pulmonary Immunology, University of Texas Health Science Center at Tyler, Tyler, TX 75708, USA; 3Beltala Barasat, 38/21, Chowdhury Para Road, Kolkata 700124, India

**Keywords:** Lactic acid bacteria, relative abundance, *Lactobacillus*, parasite infections, dysbiosis, gut microbiota

## Abstract

Tapeworm infection continues to be an important cause of morbidity worldwide. Recent metagenomics studies have established a link between gut microbiota and parasite infection. The identification of gut probiotics is of foremost importance to explore its relationship and function with the parasite in the host. In this study, the gut content of hosts infected with tapeworm *Hymenolepis diminuta* and non-infected host gut were disected out to determine their Lactic acid bacterial (LAB) population in MRS agar and microbial community was analysed by metagenomics. The bacterial count was calculated on a bacterial counting chamber and their morphology was determined microscopically and biochemically. Further, to determine the safety profile antibiotic resistance test, antimicrobial, hemolytic activity, and adhesion capability were calculated. We found six dominant probiotic strains and a decrease in LAB load from 1.7–2.3 × 10^7^ CFU/mL in the uninfected group to a range of 8.4 × 10^5^ CFU/mL to 3.2 × 10^5^ CFU/mL in the infected groups with respect to an increase in the parasite number from 10–18. In addition, we found a depletion in the probiotic relative abundance of *Lactobacillus* and an enrichment in potentially pathogenic *Proteobacteria*, *Fusobacteria*, and *Streptococcus*. Phylogenetic analysis of the six probiotics revealed a close similarity with different strains of *L. brevis*, *L. johnsonii*, *L. taiwansis*, *L. reuteri*, *L. plantarum*, and *L. pentosus*. Thus, this study suggests that the parasite inhibits probiotic colonization in the gut during its early establishment of infection inside the host.

## 1. Introduction

Tapeworms (Cestoda) continue to be an important cause of morbidity globally. Hymenolepiasis is a tapeworm infection caused by parasites belonging to the genus *Hymenolepis*. *H. nana* the dwarf tapeworm and *H. diminuta* the rat tapeworm are both armed with hooks around their four suckers; the former frequently causes human infections–being its definitive host–whereas the latter exclusively cause infections in rats and rarely infects humans, though both can infect man and rat. Most *Hymenolepis* spp. infection in human remains asymptomatic, while in heavy infestations, individuals may display common clinical symptoms such as diarrhea, abdominal pain, and vomiting [1,2]. Infection with this parasite during pregnancy may have an impact on both the health of the mother and the neonate [3]. Therapeutic prenatal interventions have impacting health issues for both the mother and the neonate, with the latter potentially developing the risk of metabolic disorders [4,5]. While morbidity can be relatively high, mortality is often modest in healthy hosts but frequently life-threatening in people with incompetent immune systems. Mental and development retardation in children is also a big problem with this infection [6]. Helminth infection is common in developing countries with tropical climates–such as India–where there is a high population, owing to the simple transmission from one person to another [5]. There are frequent reports throughout the world of intestinal parasites in adult, including pregnant women, despite the fact that the majority of these infections occur among school-age children [7]. The use of anthelmintic drugs on infected individuals is the most adopted strategy to manage tapeworm infection. However, despite widespread use of such available drugs, it is evidently difficult to reduce the prevalence of infection [8]. This problem is mostly due to inadequate hygiene and environmental contamination that leads to reinfection, and to the use of repeated commercially available anthelminthics for therapy leading to the development of drug resistance [9,10]; therefore, new strategies to control this parasite are an urgent need.

Over and above the parasites, the human gut also represents a complex ecosystem with a vast microbial population (Human Microbiome Project Consortium, 2012). Approximately 1013–1014 bacteria inhabit the gastrointestinal tract; together, they are known as the gut microbiota, and they carry out a variety of specialized roles for the host such as the absorption of nutrients, biosynthesis of vital organic substances, protection, and support of the gut immune system [11,12]. Dysbiosis has been implicated in many diseases such as cancer, type I and type II diabetes, obesity, malnutrition, and neurological problems [11,13]; thus, probiotics have become highly important for this purpose. Probiotics are live, non-pathogenic microorganisms that promote a healthy balance in the gut microbiota and upregulation of the immune system, which helps in the prevention and treatment of pathogens [14,15]. The genus *Lactobacillus* stands out among many bacteria with probiotic properties since they can resist an acidic pH and may thus survive transit through the GI tract [14]. Additionally, these microbes have the ability to prevent or reduce the colonization of pathogens in the digestive tract [15,16], which may be considered a possible weapon for the better management of numerous infectious processes, including tapeworm infection. The indigenous gut microbiota significantly contributes to the growth, health, and development of the host [16], protecting against infections in the gut, and supporting the host’s digestion by generating exogenous vitamins and digestive enzymes [17]. The use of these beneficial microorganisms as an alternative to antimicrobial substances for the prevention and control of parasitic infections has become a hot topic for research. It has been shown that parasites affect a range of host activities, including eating and reproduction, and such changes are influenced by physiological mechanisms including neurotransmitters and hormones [18]. Thus, it is possible to interpret the behavioral alterations of the host as a result of a complicated communications network between macro- and microorganisms. Specific bacterial populations should have an impact on the establishment of parasites, and parasites have the ability to alter the diversity of the bacterial microbiota. This mutual competition for the same nutrient or ecological niche makes this two-way relationship possible. It is also possible if the presence of parasites triggers the host’s immune response, upsetting the various homeostatic interactions between the host and bacterial microbiota. It has been shown that the gut microbiota retains some ecological stability, even if an individual community may contain a large variety of microbial species. This characteristic ensures that the favourable symbionts and their functions are maintained throughout time, which is eventually crucial for host health and wellbeing [19]. Additionally, the barrier effect of the intestinal epithelium by the commensal microbiota protects the host and serves as an obstacle to pathogen invasion [20]. The balance between host and gut microbiota is altered by intestinal parasites’ interactions with the microbial community, and each of these species interact with one another to metabolize and modify substrates. Numerous microorganisms of the human intestine, including *Lactobacillus*, have often been used as feed additives to investigate their impacts on growth performance, immunity and development, and dietary superiority. Moreover, since its population density regulates various animal–pathogen interactions–a process termed as quorum sensing–it has become even more unique to use it as a medication, bio-control agent, and an alternative for controlling the intestinal parasite population [21,22,23,24,25]. These potential qualities may be effectively used to treat infections in humans as well. Probiotics are often chosen based on their growth and colonization in the gastrointestinal mucus, adhesion, or in vitro antagonist activity [16]. The survival and physiology of many parasites, as well as the outcome of many parasitic diseases, may be significantly impacted by resident microbiota products. On the other hand, both intestinal protozoans and helminth parasites continuously excrete and release chemicals that may modify the environment and change the makeup of the gut microbiota. Moreover, part of the energy derived from the nutrition metabolism by inhabitant bacteria may be advantageous to both the host and parasites [26]. Therefore, it is important to consider the intestinal environment as an ecosystem where interactions between the host, parasites, and microbial populations happen at different organizational levels. 

Thus, the present study aimed at understanding the relationship between *H. diminuta* and the gut *Lactobacillus* during the parasite development. Metagenomic analysis renders an unprecedented opportunity to probe the reaction and alteration of the microbial communities in the gut environment. We report the microbial community structure in an infected rodent as revealed through metagenomic analysis sequencing method.

## 2. Materials and Method

### 2.1. Maintenance of the Parasite

*H. diminuta* was maintained in our laboratory between two hosts, the Swiss Albino rat as the definitive host and the insect beetle *Tribolium* sp. as the intermediate host, following the method as described by Kundu et al., 2012 [27]. The required animals were approved by the Institutional Animal Ethics Committee (IAEC) of Visva-Bharati University with reference no. VB/CPCSEA/IAEC/II-09/SEPT, 2019.

### 2.2. Sample Collection and Determination of Bacterial Load

Rats were divided into five groups, each having six numbers, of which Group A was the control with no infection while Group B-F were infected with 10, 12, 14, 16, and 18 numbers of the larval stage of the parasite cysticercoid respectively. All rats were fed with standard mice feed and water ad libitum. Three weeks post inoculation, the infected rats were sacrificed, their intestines were dissected out, and developed adult parasites were counted. About 1 gm of the gut content was aseptically inoculated on Mann-Rogosa-Sharpe (MRS) agar to recover pure culture [28]. From a single colony obtained, broth cultures were created and determined the Lactic acid bacteria (LAB). Another set of 1 gm content was collected from the control and infected rats for metagenomic analysis. LAB was determined in the form of Colony Forming Units (CFU/mL) as adopted from Mathialagan et al., 2018 [29]. The colonies were counted by the following formula:Colony Forming Units (CFU) = [Number of colonies × dilution factor] per mL of culture,

The differences between the mean bacterial load in the control and parasite-treated rats were tested statistically by ANOVA and Tukey’s test to determine the significant differences.

### 2.3. Metagenomic Sequencing of the Gut Sample

In order to study the diversity of bacteria, metagenomic analysis was performed in which the gut pool content from two separate samples of rats were collected, i.e., non-infected or control rats with three replicates–Control 1, Control 2, and Control 3–and *H. diminuta*-infected rats (with a standard amount of 12 cysticercoid) also with three replicates–Infected 1, Infected 2, and Infected 3–washed in saline three times for the removal of any attached bacteria from the gut wall. The DNA was then extracted using a QIAGEN extracting kit and by following the manufacturer’s protocol. Extracted DNA was amplified in PCR with forward primer V13F 5′AGAGTTTGATGMTGGCTCAG3′ and reverse primer V13R 5′TTACCGCGGCMGCSGGCAC3′ targeting a single gene 16SrDNA of bacteria at an annealing temperature of 60 °C. The product was purified and analysed using Ampure beads and quantitated using a Qubit dsDNA High Sensitivity assay kit. Sequencing was carried out using Illumina Miseq with a 2x300PE v3 sequencing kit. The resulting sequence was analysed using the tools of FASTQC and MULTIQC, followed by quality reads by TRIMGALORE tools. The relative abundance of the bacteria was calculated and estimated by QIIME2, MOTHUR, KRAKEN, and BRACKEN software. Taxonomic assignments were performed using SILVA, GREENGENES, and NCBI. Downstream analysis of the aligned data was performed using Microbiome-Analyst (https://microbiomeanalyst.ca, accessed on 5 June 2022), an online platform for microbiome data analysis using default parameters.

Data input was filtered, and the alpha diversity was measured and resulted with four methods like Chao1, Shannon, Simpson, and Fisher, with the statistical method of *T*-test /ANOVA. Beta diversity was also constructed at the taxonomic level of the Genus with the Bray-Curtis index distance method based on the Permutational MANOVA (PERMANOVA) statistical method and PCA plot. Further heatmaps were constructed for microbial analysis with a detailed view mode of <1500 features. The samples were clustered using a Ward cluster algorithm based on the Euclidean distance measure.

Box plots were performed to study the comparative analysis of LAB between the control and infected host gut samples at the taxa level of order Lactobacillales, family Lactobacillaceae, and genus *Lactobacillus*. All statistical tests of related box plots were performed and produced in R software.

Since *Lactobacillus* was negligible in the infected sample–a significant feature differential abundance with regards to the class taxa level–an LEfSe algorithm was developed and then evaluated by Linear Discriminant Analysis (LDA) to give a clear knowledge about the taxon that is highly present in one group while the same taxon is lowly observed in the other group.

### 2.4. Isolation and Purification of LAB

Isolation for LAB was aseptically inoculated on MRS broth (Hi-Media, Mumbai, India) culture; CaCO_3_ was added to the medium to induce more LAB growth and incubated at 37 °C for 48 h. Morphologically distinct and well-isolated colonies were selected and moved to fresh MRS agar (Hi-Media, Mumbai, India) plates by streaking to obtain pure colonies, and were maintained on glycerol stocks. Single colonies from these plates were isolated and again separated and inoculate in fresh MRS broth for probiotic confirmation, while one more set was used for colony morphology and another set was maintained for LAB culture stock until further use.

### 2.5. Confirmatory Tests of the Isolated LAB Colony

#### 2.5.1. Potassium Hydroxide (KOH) Test 

To assess the gram reaction of LAB isolates, the KOH test was performed with the protocol of Powers, 1995 [30]. In brief, LAB cultures were grown for 48 h at 37 °C on MRS agar. On a clean slide, one drop of 3% aqueous potassium hydroxide was applied, and pure colony was added to it using a sterile loop and thoroughly mixed.

#### 2.5.2. Catalase Test

This test was adopted from Mathialagan et al., 2018 [29]. Overnight cultures of pure colony were taken on a sterilized microscopic glass slide containing two drops of 3% H_2_O_2_ and the reaction was observed.

#### 2.5.3. Spore Staining

LAB isolates were taken for the spore-staining procedure as per Prescott et al., 2002 [31]. The isolates were then observed under light microscopy. The isolates producing no endospores were selected and further processed for the probiotic confirmation test.

#### 2.5.4. Characterization of Probiotic Properties through the In-Vitro Process

Growth at Different pH Level: The determination of the pH tolerance of isolated bacteria was determined following the adopted protocol from Liong and Shah, 2005 [32].Bile Salt Tolerance Test: This was determined following the method adopted from Liong and Shah, 2005 [32]. In brief, 2% Bile salt (Oxgall, Hi-Media, Mumbai, India) was mixed with MRS broth in test tubes. Each test tube was inoculated with 100 µL fresh culture of isolated LAB and incubated at 37 °C for 48 h, and the growth was observed.Curd Production: For the determination of the Curd production test of isolated bacteria, the standard 4% of bacteria culture was added to 5 ml of boiled and cooled milk, incubated for 24 h, and then observed.

Probiotics were identified based on their physiological, morphological, and biochemical characteristics according to Bergey’s Manual [33]. 

#### 2.5.5. Morphology of LAB Bacteria

For the identification of the cell shape and arrangements, the isolated LABs were first heat-fixed on a sterilized microscopic slide and then analysed for the gram staining test and observed under light microscopy.

#### 2.5.6. Growth at Different NaCl Concentration

The LAB isolates were determined for NaCl concentration following the method adopted from Chowdhury et al., 2012 [34]. 

#### 2.5.7. Antimicrobial Activity, Hemolytic Activity and Cell Adhesion Property 

To evaluate the safety profile of LAB isolates, antimicrobial activity tests against several pathogenic bacteria were performed following the method adopted from Jung et al., 2021 [35]. Hemolytic activity was calculated by streaking a blood agar plate [36], and the cell adhesion property was also determined by observing under a light microscope.

#### 2.5.8. Antibiotic Susceptibility Test

LAB strains of approximately 10^8^ CFU were subjected to eight antibiotics in MRS agar plates and incubated at 37 °C for 24 h. The antibiotics used were Ampicillin(AMP)-10 mcg/disc, Methicillin(MET)-5 mcg/disc, Trimethoprim(TR)-5 mcg/disc, Amoxyclav(AMC)-30 mcg/disc, Polymyxin B(PB)-300 units/disc, Penicillin G(P)-10 units/disc, Erythromycin(E)-10 mcg/disc, and Rifampicin(RIF)-5 mcg/disc obtained from Hi-Media, Mumbai, India. The zone diameter inhibition (ZDI) values were measured and interpreted according to the CLSI criteria following the method adopted from [37,38].

### 2.6. Phylogenetic Analysis of Probiotic LAB Isolates Using 16S rRNA Sequencing

The LAB isolates that passed the probiotic confirmatory test were then released and washed two times in saline containing 0.1% Tween 80 for 30 s in each wash and then centrifuged at 27,000 Ug for 20 min. DNA was then extracted using phenol/chloroform extraction and ethanol precipitation [39]. Next, 16S rDNA genes were amplified in PCR using eubacterial primer 16 forward primer 395 (5′AGAGTTTGATCMTGGCTCAG3′) and the 16S Reverse Primer 396 (3′ TACGGYTACCTTGTTAACGACTT5′). The PCR reaction mixture was the same as described above. Next, denaturation at 95 °C for 15 s, followed by annealing at 60 °C for 15 s, and elongation was done at 72 °C for 2 min, final extension at 72 °C for 10 min, and hold at 4 °C. The PCR product was then purified as mentioned above and partially sequenced using FASTQC and MULTIQC tools, followed by quality reads by TRIMGALORE tools. Partial 16S rDNA sequences were compared to the GenBank database NCBI, with the BLAST software. Clustal W, a tool for multiple alignment, was used to analyse the alignments of the sequences, and MEGA 7 was conducted to generate the phylogenetic tree.

## 3. Result

### 3.1. Determination of LAB Load

The bacterial load from parasite-treated rats significantly differ from each other. However, there is no significant difference between bacterial load in the control rats. Group A showed LAB load of 1.7–2.3 × 10^7^ CFU/mL while Group B showed 8.4 × 10^5^ CFU/mL, followed by Group C with 7.2 × 10^5^ CFU/mL, Group D with 6.4 × 10^5^ CFU/mL, Group E with 5.6 × 10^5^ CFU/mL, and Group F with 3.2 × 10^5^ CFU/mL. There is an honest negative difference in the bacterial load with increasing parasitic load, i.e., the bacterial load decreases significantly with increasing parasitic load (Figure 1a). The differences in mean bacterial load for 10–12, 12–14, and 14–16 are much smaller than the difference for 16–18. The gradual increasing parasitic load from 10 to 16 causes a significant loss in bacterial load. The increment in parasitic infection after 14 causes more loss in bacterial load. The 10–12 parasite infection increment causes a more significant loss in the bacterial difference than the 12–14 and 14–16 (Figure 1b). The LAB load in the case of 18 numbers of parasite infection shows a potential outlier (Appendix A).

### 3.2. Metagenomic Analysis

Multiple sequencing of amplicons of 16S rDNA were analysed from two groups of gut microbial communities–three times each. The Operational taxonomic unit (OTU) gives an overall microbial community present in the host gut of two groups: the control and *H. diminuta*-infected. The relative abundance of various phyla, classes, orders, families, and genera was represented by histograms and depicted in Figure 2. A total of four major bacterial phyla were in the control gut and five were found in the infected host gut. Bacterial phyla that were present in the two host guts were Fusobacteriota, Proteobacteria, Bacteroidota, Firmicutes, and Actinobacteriota, while a large proportion of reads in two samples were classified as others since no taxonomic rank lower than the domain could be ascribed to them. Actinobacterriota that was missing in the control was present in the infected sample. While Fusobacteriota and Proteobacteria were abundant phyla in the control sample with 35% and 29% respectively, Proteobacteria was the exclusively most dominant phyla in the infected sample with 90% abundance (Figure 2a). In the two samples, different bacterial classes were present in variable proportions. Fusobacteria, Gammaproteobacteria, Bacteroidia, Bacilli, Alphaproteobacteria, and Actinobacteria were the most prominent classes observed, while a large proportion of reads in two samples could not be assigned any taxonomic rank below the domain and were labeled as others. While Alphaproteobacteria and Actinobacteria were missing in the control sample, Bacteroidia and Bacilli were missing from the infected sample. Gammaproteobacteria constitute the abundant class in the infected sample with 68%, followed by Alphaproteobacteria (27%), while in control sample, Gammaproteobacteria was prominent, but with a low relative abundance (23%) (Figure 2b).

About 24 families were observed in our study, of which 11 families were present in the control sample and 13 families were present in the infected sample. The major families present were Fusobactriaceae, Bacteroidaceae, Morganellaceae, Enterococcaceae, Enterobacteriaceae, Rhizobiaceae, Moraxellaceae, Lactobacillaceae, Pseudomonadaceae, and Staphylococcaceae, and a small proportion of reads in two samples could not be assigned any taxonomic rank below the domain and were labeled as others. While Pseudomonadaceae and Staphylococcaceae were absent from the control sample, they however showed their presence in the infected sample with 3.5% and 1%, respectively, and the rest of the major families were present in both samples (Figure 2c). A total of 34 major Genus were present in this study. In the control sample, *Fusobacterium* showed high abundance with 36%, followed by *Bacteroides* (22%), *Phyllobacterium* (7%), *Enterococcus* (10%), *Enterobacter* (6%), *Morganella* (6%), *Proteus* (4%), and *Lactobacillus* (1%), while in the infected rat, *Escherichia Shigella* (41%), *Phyllobacterium* (24%), *Morganella* (18%), *Proteus* (5%), *Pseudomonas* (5%), *Rhodococcus* (2%), *Acinetobacter* (1.5%), *Staphylococcu* (1.3%), and 2% of others were present, indicating the presence of more pathogenic and opportunistic bacteria such as *Escherichia Shigella*, *Pseudomonas*, and *Staphyllococcus* (Figure 2d). Furthermore, heatmap analysis showed a correlation coefficient between the control and infected samples, as depicted in Figure 3. 

*Fusobacterium* dominated the highest position amongst the top 10 enriched bacteria, while the probiotic *Lactobacillus* showed 1% in the control host gut (Figure 4a). However, in the infected host, *Morganella* dominated the top 10 enriched bacteria with 51%, while *Lactobacillus* was negligible (Figure 4b). 

LDA study showed that some taxon such as *Lactobacillus*, *Enterococcus*, *Bacteroides*, and *Fusobacterium* were found to be present in the control sample, but the same was negligible in the infected group and vice versa (Figure 4c).

The gut microbiome alpha diversity increased in the infected samples from that of the control, as depicted in Figure 5. The *p*-value in Chao1 alpha diversity was measured to be 5.0164 × 10^−5^ with the ANOVA F-value of −18.762 (Figure 5a), in Shannon alpha diversity was measured to be 9.106 × 10^−5^ with the ANOVA F-value of −43.812 (Figure 5b), in Simpson alpha diversity was measured to be 0.0013357 with the ANOVA F-value of −8.1071 (Figure 5c), and in Fisher alpha diversity the *p*-value was 3.3916 × 10^−5^ with the ANOVA F-value of −20.518 (Figure 5d). 

The gut microbiome beta diversity shift in the infected group (Figure 6a) and the *p*-value was <0.1 with the PERMANOVA f-value of 6347.9, as depicted in (Figure 6b).

Comparative analysis between bacterial counts (filtered and log transformed) in control and infected hosts at different taxonomic levels was depicted in Figure 7. There is an observable difference between control and infected counts at the level of the Order Lactobacillales for both the filtered value and the log transformed value. The Welch *t* test confirms that this difference is significant at *p*-value = 0.09682 (for filtered) and *p*-value = 0.007957 (for Log transformed counts) (Figure 7a). Further, at the level of the Family Lactobacillaceae, there is again an observable difference between the bacterial loads (filtered and log transformed counts) in control and infected group. These differences are again confirmed to be significant at *p*-value = 0.1713 and *p*-value = 0.01579 for filtered and log transformed count (Figure 7b). Figure 7c indicates there is no major observable difference between the filtered counts in infected and control samples, but the difference between logarithmic values is observed. The Welch t test confirms the insignificance in difference between the filtered counts at genus level (*p*-value = 0.1713). The difference between the logarithmic values of genus counts is significant at *p*-value = 0.01579 as per the test.

### 3.3. Isolation and Purification of LAB

A total of 29 hemispherical white or achromatic colonies viz S1–S20 and S24–S29 were obtained from non-infected gut while S21, S22, and S23 were obtained from parasite-infected gut. 

### 3.4. Probiotic Confirmation Tests of LAB Isolates

Out of 29 LAB strains, 28 were gram positive (KOH negative) and rod shaped, while 27 were catalase negative and all strain were spore staining negative (Table 1). However, only 8 selected strains were pH tolerant, 8 strains were bile salt tolerant, and 7 strains qualified for curd production positive. Out of seven strains, one strain S1 failed to grow at different pH level, while the other six strains showed their growth in 48 h of incubation and also survived in different NaCl concentrations. All the LAB isolates showed γ hemolytic pattern in blood agar plates. After being exposed to antibiotic susceptibility (Table 2), antimicrobial tests, and cell adhesion properties, the six strains were finally confirmed as probiotics, S10, S13, S14, S17, S27, and S29. All six of these probiotic strains were found in uninfected rat. Further, these six strains were subjected to 16S rDNA phylogenetic analysis.

### 3.5. Phylogenetic Analysis

We assessed the degree of similarity of the six LAB strains based on the degree where two communities share a branch length of a master phylogenetic tree constructed from two samples obtained in our study. The data produced from six strains of *Lactobacillus* were used to calculate genetic relatedness, and by pairwise comparisons, the dendogram was constructed (Figure 8). The sequences of *Lactobacillus* species were aligned by the computer and compared with the sequences representing the genera *Lactobacillus*. Homology values were determined, and the presumptive relationships of these sequences were obtained from database comparison. As shown in Figure 8a, S10 showed a 99.37% or higher identity to the database sequences with the closest species of *Lactobacillus brevis* ATCC 14869 = DSM 20054 (Appendix A), S13 showed 88.46% similarity to *Lactobacillus taiwanensis* strain BCRC 17755, and *Lactobacillus johnsonii* strain CIP 103620 (Figure 8b, Appendix A); S14 showed 99.85% similarity to *Lactobacillus johnsonii* strain 1696, 1000, 681, 680, 667, 666, 658, 1447, 1029, and 7991 (Figure 8c, Appendix A); S17 showed 99.72% similarity with *Lactobacillus reuteri* DSM 20016 (Figure 8d, Appendix A); S27 showed 96.81% similarity to *Lactiplantibacillus plantarum* strain CIP 103151, JCM 1149, NRRL B-14768, and *Lactobacillus pentosus* strain 124-2 (Figure 8e, Appendix A); and S29 showed 99.87% homology with *Lactobacillus johnsonii* strain 103620 (Figure 8f, Appendix A).

## 4. Discussion

The manipulation of human gut microbiota was suggested to improve the consequences caused by helminth infections [40]. However, extensive studies regarding the mechanisms of helminth infection/pathology interaction with the gut microbiome are required. In our study, Swiss Albino rats, being the definitive host for *H. diminuta*, serve as a great animal model for study on the alterations of LAB during this tapeworm infection.

In the present study, we reported that with increase in the parasite number, there was a decrease in LAB load. This suggests that the alteration in the *Lactobacillus* population was partly dose-dependent. A similar observation was also reported in the gut microbiota of mice exposed to mycotoxin and deoxynivalenol [41]. Intestinal helminths have been reported to alter mucosal secretion, intestinal physiology, permeability, and antimicrobial peptide production, all of which may impact the gut microbiome, including probiotics [42]. The 10–12 number of parasite infection decreased the LAB load more significantly than the 12–14 and 14–16 infection. Correspondingly, the overall LAB load remained constant as these changes occurred in their gut microbial ecology. This trend in differences indicated that the stress level due to infection faces satiation for further parasite invasion until the parasite number increases over 16. Similar findings were seen in *Microtus fortis* infected with *Schistosoma*, where the microbiome composition declined in the early stages of infection and a progressive recovery in response to the host’s natural resistance mechanism [43]. 

The alteration of the taxonomic structure was seen in the intestinal microbial profiles of Albino rat infected with *H. diminuta* from the uninfected rat. We found that Fusobacteriota, Proteobacteria, Bacteroidota, and Firmicutes are the most abundant phyla in the digestive system of the control rat, which accounts for up to 97.5%, while in the infected rat, the phylum Proteobacteria alone comprises 90% of the total microbiome abundance. Similar observations were reported in mice with induced obesity [44], in children with a low-fat, high-fiber diet [45], and in the snail *Biomphalaria* sp. infected with *Schistosoma mansoni* [46]. The increased Proteobacteria suggested an unstable microbial population (dysbiosis), which is a potential diagnostic criterion for infection or disease [47]. The Genera *Fusobacterium* and *Bacteroides* were mainly abundant in the control sample and the probiotic *Lactobacillus* was 1%, but in the infected sample *Lactobacillus* was negligible. Similar observations were reported from caecal intestine and faecal samples of rat infected with *H. diminuta* [48,49] and in canine infected with *Giardia intestinalis*, where a reduction in *L. jhonsonii* was observed [50]. Our results are contrary to the findings of Kim et al., 2019 [51] where the relative abundance of *Lactobacillus* increased with *Clonorchis sinensis* infection. This might suggest a direct interaction of excretory secretory products of the intestinal parasite *H. diminuta* preventing the colonization of the probiotic population at the early stage of infection. Our study also showed a large diversity of enterobacteria species in the infected sample. A similar observation was seen in rodents infected with nematode infection [52]. Several studies reported that pathogenic and opportunistic bacteria showed their presence in many infections where probiotics were altered, which is generally attributed to a preferential overgrowth of the Gram negative enterobacteria, although there could be a multifactorial etiology of intestinal disruption [53,54]. 

We also revealed that out of 29 isolates, only six LAB confirmed their probiotic characters. They showed acidic pH tolerance, inferring that these strains would probably survive in the stomach’s acidic environment. Our findings are consistent with those from earlier research [55,56,57]. Additionally, bile tolerance is one of the important aspects needed for LAB to live and perform as intended in the small intestine. The efficacy of bile’s inhibitory effects is mostly determined by the concentration of bile salts, which play a crucial role in both specialized and general defensive mechanisms in the gut [58]. Therefore, probiotic bacteria should be able to tolerate 0.15–2% bile. Our study also showed that the six strains of *Lactobacillus* possessed tolerance against the bile salts. For food industrial applications, LAB’s salt tolerance is a significant factor. Loss of turgor pressure in cells due to high salt concentrations may have an impact on their physiology, water activity, and enzymatic activity [59]. Lactic acid is created by LAB during fermentation operation, hence alkaline solution is added to avoid acidic conditions and raising the osmotic pressure of the cell [60], which is similar to our findings. We evaluated the antibiotic resistance profile to make sure that none of the selected probiotic strains included genes that may transmit antibiotic resistance, as this may act as a gene repository for antibiotic resistance [41]. By genetically transferring to non-resistant bacteria, this reservoir of antibiotic resistance genes may contribute to the spread of multi-antibiotic resistant bacteria [61]. All the tested antibiotic strains were susceptible or intermediately susceptible to eight tested antibiotics, which support the safety of these probiotic strains. Furthermore, all six LAB isolates exhibited γ-hemolytic activity and are thus approved for human consumption [62]. These six LAB isolates’ antimicrobial resistance profile, hemolytic activity, and cell adhesion property provide some preliminary guidance for selecting the best probiotics LAB isolate for further use. 

The identification and characterisation of LABs are vital since they are used as probiotics. Due to the biochemical similarities of LABs, molecular tests are the most efficient and accurate approach for LAB characterization and differentiation [63]. Since it takes around 17 phenotypic tests to identify a LAB isolate at the species level, correct identification of LABs by phenotypic approaches is challenging. For this reason, choosing new strains of bacteria from a variety of bacterial isolates requires the identification of microorganisms that exhibit probiotic qualities with nutritional and economic relevance. *Lactobacillus* isolated from various sources have been effectively identified at the species level using restriction profiling of 16S rRNA [64]. There were a limited number of reports regarding the molecular characterization of *Lactobacillus* in helminth-infected hosts until recent years. The results of this investigation showed that rats’ gut microbiota contains a diversity of LAB. The 16S rRNA gene sequencing identified the isolates as different strains of *L. brevis*, *L.taiwanensis*, *L*. *johnsonii*, *L. reuteri*, *L*. *plantarum*, and *L. pentosus*. A similar study found, *L. taiwanensis* strain BCRC 17755 screening from the mice gut was reported to induce the expression of REGs in the epithelial cells of the [65]. The regulation of FABP4, adiponectin, and adipsin by *L. taiwanensis* strain BCRC 17755 in Paneth cells provided a clue to reveal the relationship between the metabolic syndrome and gut microbiota [66]. *L. plantarum* strain CIP 103151 isolated from few traditionally fermented food products was reported to have probiotic properties [67,68,69]. Our studies showed a low interspecies sequence variation, whereas inter strain sequence similarity was higher. The low sequence divergence levels at 16S rRNA among LABs isolated from rat intestine have been reported by other workers. Since about 50% of accessory genes code for proteins with roles in carbohydrate metabolism and transport, the low sequence divergence among LABs can be attributed to bacteria’s adaptation to a rich and variable carbohydrate content. The results of the present study revealed that the probiotics isolated from rat intestine mainly belonged to the genera *Lactobacillus*. This could be one of the most common probiotics in the gut of rodents. Similar studies from gut of human and other rodents showed *Lactobacillus* and *Bifidobacterium* spp. as the predominant probiotics [70,71].

Therefore, our study revealed that at the early stage of infection with *H. diminuta*, inflammatory response could have been elicited by the host as observed and reported by other workers [72,73]. This response cannot be neutralized if probiotics are eliminated, as these organisms are also anti-inflammatory [15]; it may instead lead to the retaining of the inflammatory response all along the maturation of the parasite. Additionally, other health benefits of probiotics might also be restricted in the early stage of *H. diminuta* infection in hosts.

## 5. Conclusions

This study provides a preliminary view of the effects of early infection by *H. diminuta* on the gut LAB as well as probiotics colonization. The results will help to understand the antagonistic relationship between LAB and *H. diminuta*, which leads to the dysbiosis of the whole microbial community in the gut. To prove if these changes in the host gut microbiota might cause a dysbiosis-related illness like irritable bowel syndrome or inflammatory bowel disease, further research is required. The absence of probiotics facilitates more colonization of Proteobacteria in the gut, which may lead to more illness along with parasite infection. Further studies on long-term parasite infection will be required to have a complete profile of probiotic strain colonization. The identified potential probiotics supplement might provide new perspectives for health benefits in humans in the future.

## Figures and Tables

**Figure 1 microorganisms-10-02328-f001:**
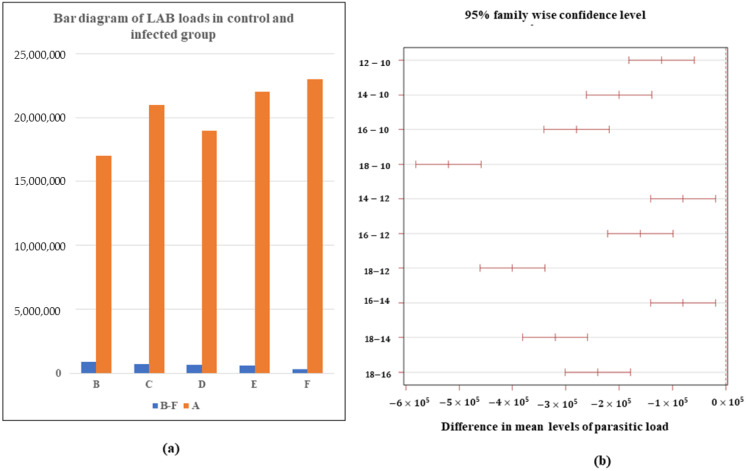
LAB load in rats: (**a**) Comparison between control (Group A) and infected rats (Group B, C, D, E and F with 10, 12, 14, 16 and 18 parasites) Each Group n = 6; (**b**) Tukey’s honest significant test showing significance of differences of LAB load between pairs of Group means (12–10, 14–10, 16–10, 18–10, 14–12, 16–12, 18–12, 16–14, 18–14, and 18–16).

**Figure 2 microorganisms-10-02328-f002:**
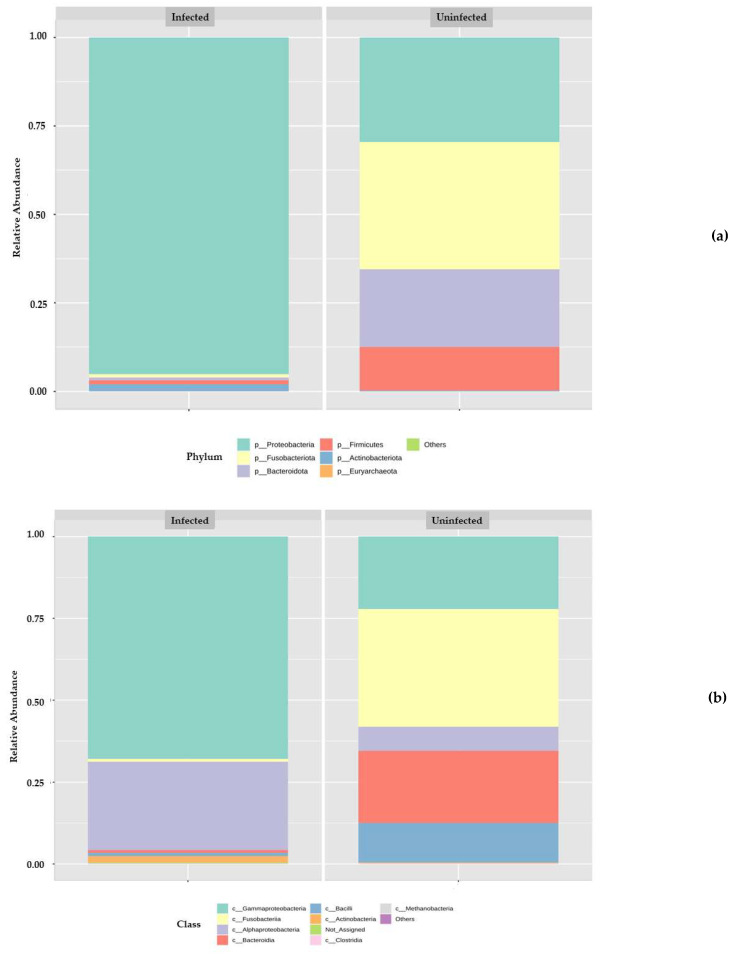
Operational taxonomic units (OTUs) presentation and analysis of micro bacteria from Phylum to Species level showing >1% relative abundance between Infected and Uninfected host gut: (**a**) Phylum level; (**b**) Class level; (**c**) Family level; (**d**) Genus level.

**Figure 3 microorganisms-10-02328-f003:**
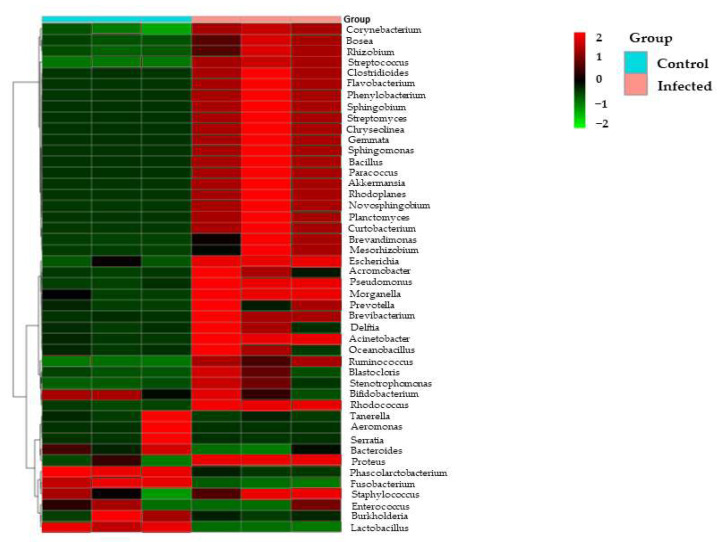
Heatmap showing the normalized relative abundances which were significantly changed in Control vs. Infected groups.

**Figure 4 microorganisms-10-02328-f004:**
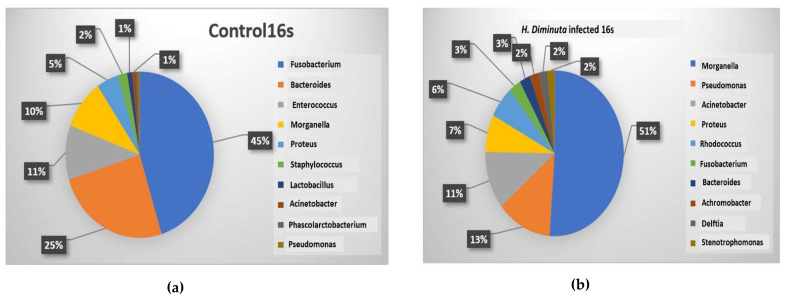
Percentage of top 10 genera of micro bacteria present in the gut of (**a**) Control and (**b**) Infected; (**c**) Linear Discriminant Analysis showing OUT maximum probability of abundance of different Genera in control and infected host gut.

**Figure 5 microorganisms-10-02328-f005:**
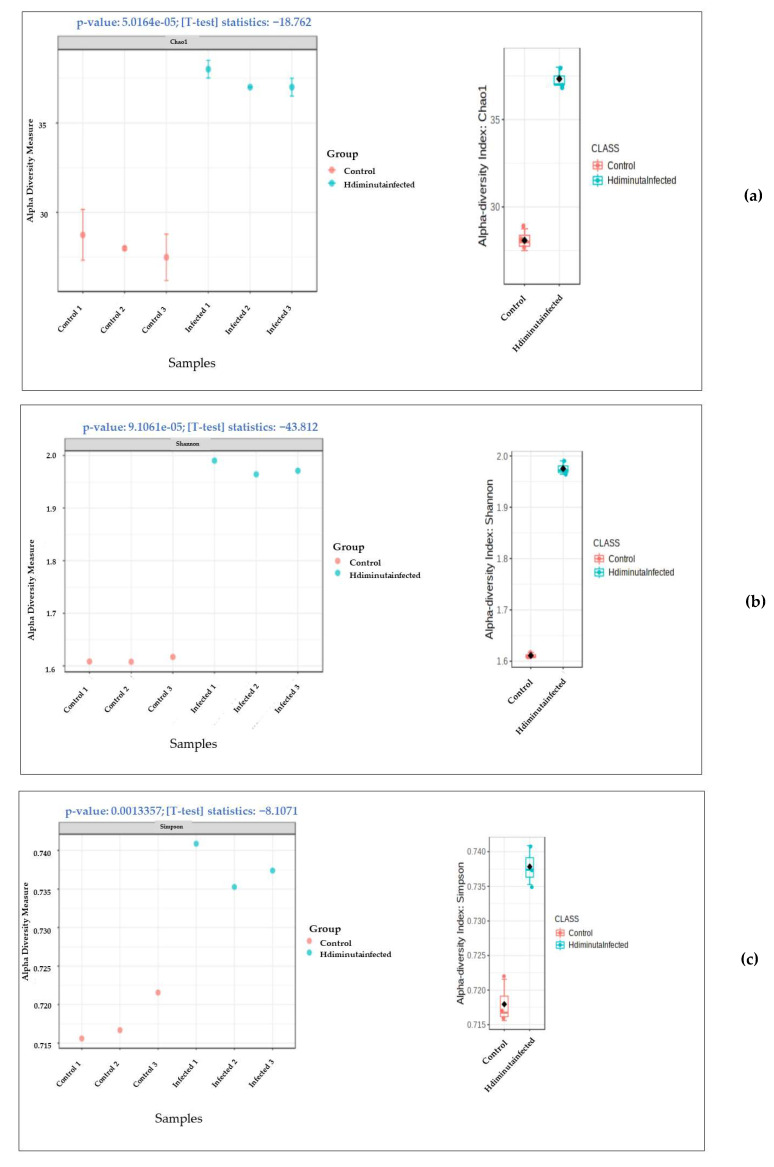
Metagenomic analysis of functional alpha diversity of gut microbiota of Control (3 replicas-Control 1, 2 and 3) and Infected (3 replicas- Infected 1, 2 and 3) samples (**a**) Chao1 alpha diversity measure and index (**b**) Shannon alpha diversity measure and index (**c**) Simpson alpha diversity measure and index (**d**) Fisher alpha diversity measure and index.

**Figure 6 microorganisms-10-02328-f006:**
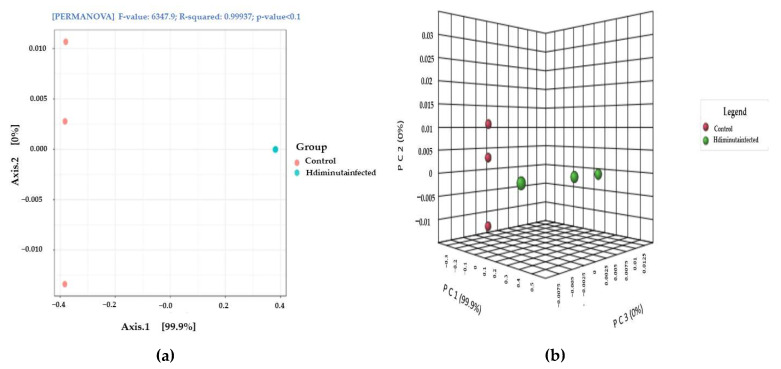
Beta diversity analysis: (**a**) PERMANOVA showing shifting of microbiome from Control to Infected; (**b**) 3-D PCA plot highlighted the shifting of microbiome.

**Figure 7 microorganisms-10-02328-f007:**
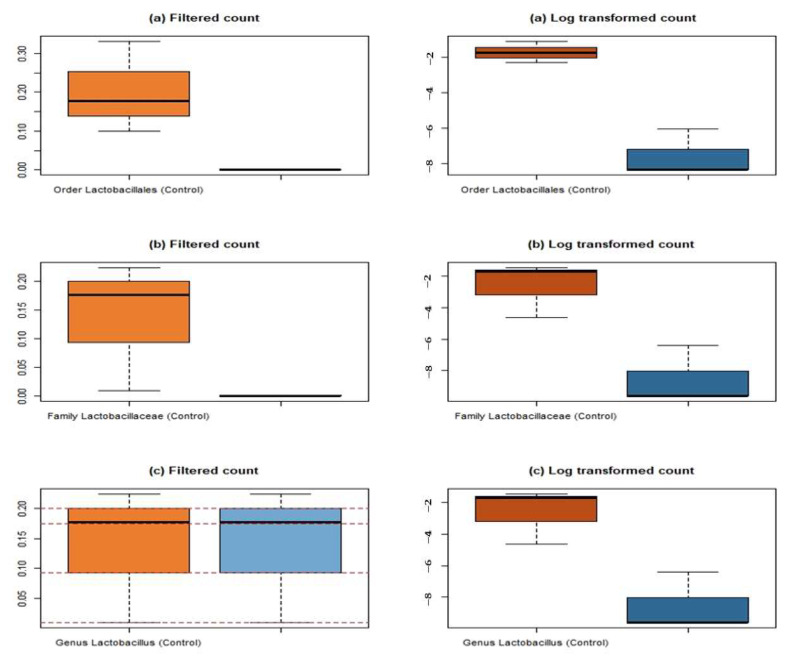
Box plot diagram for comparative count analysis of LAB showing significant variability in control and infected host gut at the taxa level: (**a**) Order Lactobacillales; (**b**) Family Lactobacillaceae; (**c**) Genus *Lactobacillus*.

**Figure 8 microorganisms-10-02328-f008:**
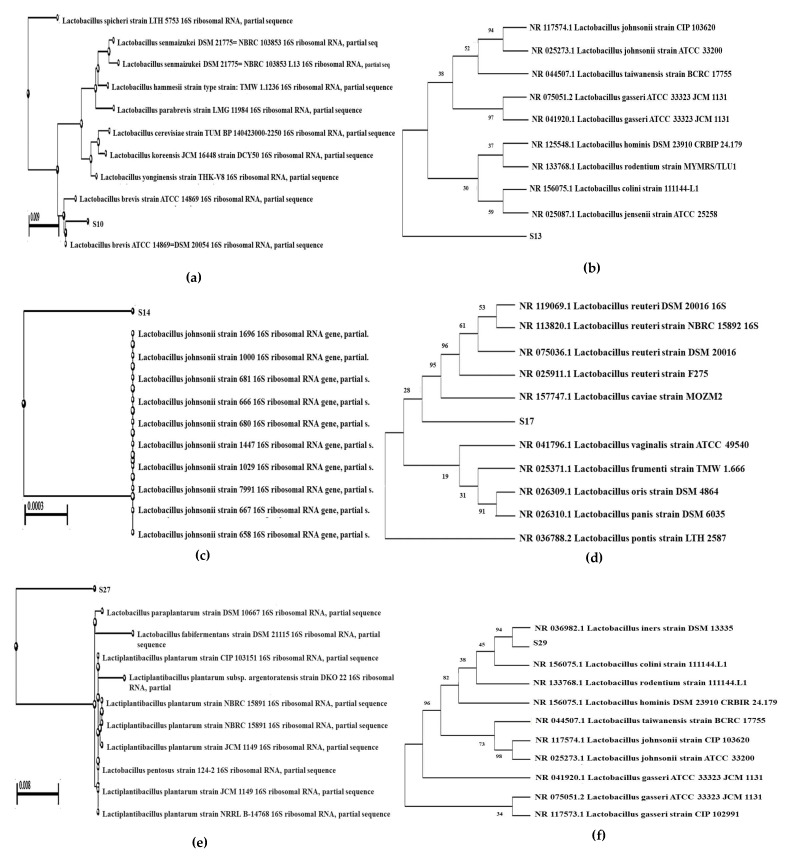
Dendogram showing the homology of six LAB strains (**a**) S10; (**b**) S13; (**c**) S14; (**d**) S17; (**e**) S27; (**f**) S29, with 51 strains representing the genera *Lactobacillus*.

**Table 1 microorganisms-10-02328-t001:** Probiotic confirmatory test of 29 isolates.

Group	Strain Code	KOH Test	Catalase Test	Spore Production	Curd Production Test	Gram Staining	Salt Tolerance (NaCl)				pH Tolerance					Bile Salt Tolerance Test	Hemolytic Activity	Antimicrobial Activity
							2%	3%	4.5%	6%	2.5	3.5	5.5	6.5	7			
**CONTROL**	**S1**	**-**	**-**	**-**	**+**	**Cocci +**	**+**	**+**		**-**	**-**	**-**	**-**	**-**	**+**	**-**	**γ**	**-**
	**S2**	**-**	**-**	**-**	**-**	**Rod +**	**+**	**+**	**-**	**-**	**+**	**+**	**+**	**+**	**+**	**+**	**γ**	**-**
	**S3**	**-**	**-**	**-**	**-**	**Rod +**	**+**	**+**	**-**	**-**	**-**	**-**	**-**	**+**	**+**	**-**	**γ**	**-**
	**S4**	**-**	**-**	**-**	**-**	**Rod +**	**+**	**+**	**-**	**-**	**-**	**-**	**-**	**+**	**+**	**-**	**γ**	**-**
	**S5**	**-**	**-**	**-**	**-**	**Rod +**	**+**	**+**	**-**	**-**	**-**	**-**	**-**	**+**	**+**	**-**	**γ**	**-**
	**S6**	**-**	**-**	**-**	**-**	**Rod +**	**+**	**+**	**-**	**-**	**-**	**-**	**-**	**+**	**+**	**-**	**γ**	**-**
	**S7**	**+**	**-**	**-**	**-**	**Rod-**	**+**	**+**	**-**	**-**	**-**	**-**	**-**	**+**	**+**	**-**	**γ**	**-**
	**S8**	**-**	**-**	**-**	**-**	**Rod +**	**+**	**+**	**-**	**-**	**-**	**-**	**-**	**+**	**+**	**+**	**γ**	**-**
	**S9**	**-**	**+**	**-**	**-**	**Rod +**	**+**	**+**	**-**	**-**	**+**	**+**	**+**	**+**	**+**	**-**	**γ**	**-**
	**S10**	**-**	**-**	**-**	**+**	**Rod +**	**+**	**+**	**+**	**+**	**+**	**+**	**+**	**+**	**+**	**+**	**γ**	**+**
	**S11**	**-**	**-**	**-**	**-**	**Rod +**	**+**	**+**	**-**	**-**	**-**	**-**	**-**	**+**	**+**	**-**	**γ**	**-**
	**S12**	**-**	**-**	**-**	**-**	**Rod +**	**+**	**+**	**-**	**-**	**-**	**-**	**-**	**+**	**+**	**-**	**γ**	**-**
	**S13**	**-**	**-**	**-**	**+**	**Rod +**	**+**	**+**	**+**	**+**	**+**	**+**	**+**	**+**	**+**	**+**	**γ**	**+**
	**S14**	**-**	**-**	**-**	**+**	**Rod +**	**+**	**+**	**+**	**+**	**+**	**+**	**+**	**+**	**+**	**+**	**γ**	**+**
	**S15**	**-**	**-**	**-**	**-**	**Rod +**	**+**	**+**	**+**	**-**	**-**	**-**	**-**	**+**	**+**	**-**	**γ**	**-**
	**S16**	**-**	**-**	**-**	**-**	**Rod +**	**-**	**-**	**-**	**-**	**-**	**-**	**-**	**+**	**+**	**-**	**γ**	**-**
	**S17**	**-**	**-**	**-**	**+**	**Rod +**	**+**	**+**	**+**	**+**	**+**	**+**	**+**	**+**	**+**	**+**	**γ**	**+**
	**S18**	**-**	**-**	**-**	**-**	**Rod +**	**+**	**+**	**-**	**-**	**-**	**-**	**-**	**+**	**+**	**-**	**γ**	**-**
	**S19**	**-**	**+**	**-**	**-**	**Rod +**	**+**	**+**	**-**	**-**	**-**	**-**	**-**	**+**	**+**	**-**	**γ**	**-**
	**S20**	**-**	**-**	**-**	**-**	**Rod +**	**+**	**+**	**-**	**-**	**-**	**-**	**-**	**+**	**+**	**-**	**γ**	**-**
	**S24**	**-**	**-**	**-**	**-**	**Rod +**	**+**	**-**	**-**	**-**	**-**	**-**	**-**	**+**	**+**	**-**	**γ**	**-**
	**S25**	**-**	**-**	**-**	**-**	**Rod +**	**+**	**+**	**-**	**-**	**-**	**-**	**-**	**+**	**+**	**-**	**γ**	**-**
	**S26**	**-**	**-**	**-**	**-**	**Rod +**	**+**	**+**	**-**	**-**	**-**	**-**	**-**	**+**	**+**	**-**	**γ**	**-**
	**S27**	**-**	**-**	**-**	**+**	**Rod +**	**+**	**+**	**+**	**+**	**+**	**+**	**+**	**+**	**+**	**+**	**γ**	**+**
	**S28**	**-**	**-**	**-**	**-**	**Rod +**	**+**	**+**	**-**	**-**	**-**	**-**	**-**	**+**	**+**	**-**	**γ**	**-**
	**S29**	**-**	**-**	**-**	**+**	**Rod +**	**+**	**+**	**+**	**+**	**+**	**+**	**+**	**+**	**+**	**+**	**γ**	**+**
**INFECTION**	**S21**	**-**	**-**	**-**	**-**	**Rod +**	**+**	**+**	**+**	**-**	**-**	**-**	**+**	**+**	**+**	**-**	**γ**	**-**
	**S22**	**-**	**-**	**-**	**-**	**Rod +**	**+**	**+**	**+**	**-**	**-**	**-**	**+**	**+**	**+**	**-**	**γ**	**-**
	**S23**	**-**	**-**	**-**	**-**	**Rod +**	**+**	**+**	**+**	**-**	**-**	**-**	**+**	**+**	**+**	**-**	**γ**	**-**

+, Positive; -, Negative; γ, Gamma hemolytic pattern.

**Table 2 microorganisms-10-02328-t002:** Antibiotic susceptibilities of six isolated LAB strains.

**Strain Code**	**ZDI (mm) of Antibiotic**							
	**AMP**	**MET**	**TR**	**AMC**	**PB**	**P**	**E**	**RIF**
S10	18 ^I^	19 ^I^	18 ^I^	21 ^S^	18 ^I^	22 ^S^	26 ^S^	21 ^S^
S13	24 ^S^	16 ^I^	22 ^S^	24 ^S^	19 ^I^	18 ^I^	17 ^I^	16 ^I^
S14	18 ^I^	16 ^I^	16 ^I^	19 ^I^	22 ^S^	26 ^S^	24 ^S^	24 ^S^
S17	19 ^I^	19 ^I^	26 ^S^	18 ^I^	19 ^I^	26 ^S^	23 ^S^	24 ^S^
S27	18 ^I^	20 ^I^	18 ^I^	19 ^I^	20 ^I^	19 ^I^	23 ^S^	24 ^S^
S29	19 ^I^	16 ^I^	16 ^I^	19 ^I^	20 ^I^	23 ^S^	22 ^S^	19 ^S^

The LAB strains were grouped into resistant (ZDI: ≤15 mm), sensitive (ZDI: ≥21 mm), or intermediately susceptible (ZDI: 16–20 mm). ^S^ = Sensitive; ^I^ = Intermediately susceptible; ^R^ = Resistance.

## Data Availability

Not applicable.

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
