# Peer review of "Hymenolepis diminuta Reduce Lactic Acid Bacterial Load and Induce Dysbiosis in the Early Infection of the Probiotic Colonization of Swiss Albino Rat"

_microorganisms, 2022, doi:10.3390/microorganisms10122328_

Round 1

Reviewer 1 Report

Sudeshna et al revealed six dominant probiotic strains, and further found depletion in the probiotic relative abundance of Lactobacillus and enrichment in potentially pathogenic Proteobacteria, Fusobacteria, and Streptococcus. Furthermore, phylogenetic analysis of the six probiotics revealed close similarity with different strain of L. brevis, L. johnsonii, L. taiwansis, L. reuteri, L. plantarum and L. pentosus. They found the tapeworm Hymenolepis diminuta infection inhibit probiotics colonization in the gut during the initial stage of infection. Although the results are interesting and could provide new insights for the parasite-microbiome studies, the metagenomic analysis, and data presentation are not exhausted. It would be good to see more bioinformatics data after revision and then considered to be published. Please find below the major concerns.

Method section. Sequencing of the Gut Sample

This part is unclear. It would be better to present more information about the bioinformatics analysis, such as which pipeline was used? Qimme2 or other tools? The alpha and beta diversity didn’t analyze in the present version of the manuscript, so I strongly recommend including these types of analysis, and PERMANOVA analysis was used to determine the statistical differences. PCA plots and heatmaps are commonly used for microbiota analysis, they should be presented in the main manuscript.

Result section. Metagenomic Analysis

Please see the comments in the methods section, and present these data in the main manuscript.

Author Response

Reviewer 1:

Comments and Suggestions for Authors

Sudeshna et al revealed six dominant probiotic strains, and further found depletion in the probiotic relative abundance of Lactobacillus and enrichment in potentially pathogenic Proteobacteria, Fusobacteria, and Streptococcus. Furthermore, phylogenetic analysis of the six probiotics revealed close similarity with different strain of L. brevis, L. johnsonii, L. taiwansis, L. reuteri, L. plantarum and L. pentosus. They found the tapeworm Hymenolepis diminuta infection inhibit probiotics colonization in the gut during the initial stage of infection. Although the results are interesting and could provide new insights for the parasite-microbiome studies, the metagenomic analysis, and data presentation are not exhausted. It would be good to see more bioinformatics data after revision and then considered to be published. Please find below the major concerns.

Response to Comment 1.

  1. The authors sincerely acknowledge the comments and suggestions. We have revised our manuscript (MS) and have elaborated the bioinformatics data and incorporated in the MS as per your suggestions.

Comment 2. Method section. Sequencing of the Gut Sample

This part is unclear. It would be better to present more information about the bioinformatics analysis, such as which pipeline was used? Qimme2 or other tools? The alpha and beta diversity didn’t analyze in the present version of the manuscript, so I strongly recommend including these types of analysis, and PERMANOVA analysis was used to determine the statistical differences. PCA plots and heatmaps are commonly used for microbiota analysis, they should be presented in the main manuscript.

Response to Comment 2.

  1. The pipeline QIMME2 was used along with other tools like KRACKEN, BRACKEN and MOTHUR for comparing.
  2. The authors have added the Alpha and Beta diversity analysis with PERMANOVA analysis to determine the statistical differences of microbial biome in control and infected samples in the revised MS as suggested.
  3. PCA as well as Heatmaps analysis were also incorporated in the revised main MS and figures as suggested.

Comment 3. Result section. Metagenomic Analysis

Please see the comments in the methods section, and present these data in the main manuscript.

Response to comment 3

As per your suggestions in the method section all data have been incorporated in the revised MS.

Reviewer 2 Report

The manuscript entitled 'Hymenolepis Diminuta Reduce Lactic Acid Bacterial Load and Induce Dysbiosis in the Probiotic Colonization During its Early Infection in Swiss Albino Rat' by Mandal. et al. investigates the effects of early infection by H. diminuta on the gut LAB as well as probiotics colonization. 

The MS needs to be edited appropriately. I recommend the author make a few minimal revisions to this essay.

Abstract

L31: What do you mean of initial stage ?

Introduction

L39-46:Please confirm whether non-proprietary nouns need italics?

L56-57:“This problem is ...  underpin this statement. what do you mean of this sentence.

L71-75, 80-81, etc.: Use reference/s to back up statement.

Materials and Method

L135-136:How they were selected? It means how these samples for metagenomic analysis representative?

All of the Figures were not clear.

In Figure 3, the sequence of a and b was out of order.

Results
L317: and in the whole MS - use the correct form of the genus name, Lactobacillus spp. etc.

Line 316 and 318: are the p value for filtered count and log transformed count

same completely?

Line 385,Whichshould be deleted.

Discussed.
L473-475:Use reference/s to back up statement. Are there corresponding data from this experiment to prove the inflammatory response caused by H. diminuta?

L474-475:“more severe inflammatory during maturation of parasite,why? How to get this conclusion?

Author Response

Reviewer 2:

Comments and Suggestions for Authors

The manuscript entitled 'Hymenolepis Diminuta Reduce Lactic Acid Bacterial Load and Induce Dysbiosis in the Probiotic Colonization During its Early Infection in Swiss Albino Rat' by Mandal. et al. investigates the effects of early infection by H. diminuta on the gut LAB as well as probiotics colonization. 

The MS needs to be edited appropriately. I recommend the author make a few minimal revisions to this essay. 

Response to Comment 1.

All authors have acknowledged your comments and suggestions and we have revised the manuscript (MS) accordingly. The whole MS has been edited wherever needed appropriately.

Comment 2. Abstract

L31: What do you mean of “initial stage” ?

Response to Comment 2.

‘Initial stage’  means the early stage of establishment of infection. We have rewritten in the abstract of the MS for clarification.

Comment 3.  Introduction

  • L39-46:Please confirm whether non-proprietary nouns need italics?

Response: Non-proprietary nouns does not need to be italicised, we have italicised the scientific generic name and species wherever is needed.

  • L56-57:“This problem is ...”  underpin this statement. what do you mean of this sentence.

Response: The authors have rewritten the statement “ The problem is………needed” as suggested.

  • L71-75, 80-81, etc.: Use reference/s to back up statement.

Response: The authors have added references to back up the statement accordingly.

Comment 4. Materials and Method

>L135-136: How they were selected? It means how these samples for metagenomic analysis representative?

Response to comment 4.

In our laboratory (Ref. 27) we have established the Hymenolepis diminuta infection in rat by inoculating 10-12 cysticercoid larval stage of H. diminuta in  rats. The present study of groupings were done to see if increasing or decreasing the number of larval forms from the established number will have an effect on the LAB load or not. Thus the five groups Group- B,C,D,E and F was obtained as infected groups as well as Group A which is a control where each group contain six rats to obtain the statistical differences.  In another set of experiment we took the standard group (10-12 parasites) with significant LAB load and pool the gut content for metagenomic analysis.

Comment 5. All of the Figures were not clear.

Response to comment 5.

All figures presented are of maximum resolution

Comment 6. In Figure 3, the sequence of a and b was out of order.

Response to Comment 6.

The authors appreciate the comments and correction has been done accordingly in the revised MS.

Comment 7.Results

>L317: and in the whole MS - use the correct form of the genus name, Lactobacillus spp. etc.

Response: The authors appreciated the comment and corrections have been done accordingly in the revised MS.

>Line 316 and 318: are ‘the p value for filtered count and log transformed count’same completely?

Response: No the filtered count and log transformed count are two separate p-values. It has been clarified in the result section of the revised MS. The effect of parasitic infection on bacterial count is not negotiable as per our finding. Since logarithmic scale linearizes the distribution of values and minor handling errors in bacterial count are minimized in logarithmic scale, therefore, the significant difference in logarithmic values should be accounted for than the insignificant difference between the filtered only counts.

>Line 385,‘Which’ should be deleted.

Response. The authors have deleted the word ‘Which’ from the Revised MS.

Comment 8 Discussed.

>L473-475:Use reference/s to back up statement. Are there corresponding data from this experiment to prove the inflammatory response caused by H. diminuta?

Response: Yes there were data to prove the inflammatory response caused by H. diminuta and the references have already been added in the original MS and were kept as it is.

>L474-475:“more severe inflammatory during maturation of parasite”,why? How to get this conclusion?

Response: Since host already illicit inflammatory response against the parasite during infection and  the probiotics have also been reported to have anti-inflammatory activity, thus eliminating the probiotic will only retain the inflammatory response.The word ‘more severe’ has been removed from the revised MS.

Reviewer 3 Report

Sudeshna Mandal and collegues presented a manuscript on the effect of Hymenolepis diminuta infection on host gut microbiota, with focusing on lactic acid bacteria population. The authors performed a detailed study, showing that the infection with the tapeworm induced dysbiosis and exploring its relation to the host health and infection progression/control.

Minor comments:

Minor revision of English language and formatting, as double spaces, small errors, and italics are found in abstract and introduction.

Author Response

Reviewer 3:

Comments and Suggestions for Authors

Sudeshna Mandal and collegues presented a manuscript on the effect of Hymenolepis diminuta infection on host gut microbiota, with focusing on lactic acid bacteria population. The authors performed a detailed study, showing that the infection with the tapeworm induced dysbiosis and exploring its relation to the host health and infection progression/control.

 Minor comments:

Minor revision of English language and formatting, as double spaces, small errors, and italics are found in abstract and introduction.

Response to comments

The authors acknowledged the comments and suggestions and we have revised the English language and formatting accordingly in the revised manuscript.

Round 2

Reviewer 1 Report

The authors have addressed all my concerns.

Author Response

Respected editor, thank you for your valuable suggestions.